# Effects of speculum lubrication on cervical smears for cervical cancer screening: A double blind randomized clinical trial

Chito P. Ilika[1], George U. Eleje[1,2], Michael E. Chiemeka[3], Frances N. Ilika[4], Joseph I. Ikechebulu[1,2], Valentine C. Ilika[5], Emmanuel O. Ugwu[6], Ifeanyichukwu J. Ofor[7], Onyecherelam M. Ogelle[1,2], Osita S. Umeononihu[1,2], Johnbosco E. Mamah[8], Chinedu L. Olisa[1,9], Chijioke O. Ezeigwe[1,2], Malarchy E. Nwankwo[1,2], Chukwuemeka J. Ofojebe[1,2], Chidinma C. Okafor[10], Onyeka C. Ekwebene[11], Obinna K. Nnabuchi[1], Chigozie G. Okafor[1] *

1 Department of Obstetrics and Gynaecology, Nnamdi Azikiwe University Teaching Hospital, Nnewi, Anambra State, Nigeria, 2 Department of Obstetrics and Gynaecology, Nnamdi Azikiwe University, Awka, Anambra State, Nigeria, 3 Department of Anatomic Pathology and Forensic Medicine, Nnamdi Azikiwe University Teaching Hospital, Nnewi, Anambra State, Nigeria, 4 Health Policy Plus (HP+), Abuja, Nigeria, 5 Department of Internal medicine, Nnamdi Azikiwe University Teaching Hospital, Nnewi, Anambra State, Nigeria, 6 Department of Obstetrics and Gynaecology, College of Medicine, University of Nigeria Teaching Hospital, Enugu, Enugu State, Nigeria, 7 Department of Obstetrics and Gynaecology, College of Medicine, Enugu State University Teaching, Parklane, Enugu State, Nigeria, 8 Alex-Ekwueme Federal University Teaching Hospital, Abakaliki, Nigeria, 9 Department of Pharmacology and Therapeutics, Nnamdi Azikiwe University, Awka, Anambra State, Nigeria, 10 Department of Psychiatry, Leicestershire Partnership NHS Trust, Leicester, United Kingdom, 11 Department of Biostatistics and Epidemiology, East Tennessee State University, Johnson City, TN, United States of America

* chigolz@yahoo.com

## Abstract

### Background

Speculum lubrication may help to reduce the pain experienced during Pap-smear collection and hence increase uptake of cervical cancer screening and repeat testing, but there are fears of its interference with cytological results.

### Aim

To determine and compare the adequacy of cervical cytology smears and the mean pain scores of women undergoing cervical cancer screening with or without speculum lubrication.

### Methods

This was a randomised controlled study of 132 women having cervical cancer screening at a tertiary hospital in Nigeria. Sixty-six participants were randomly assigned to the 'Gel' and 'No Gel' groups, respectively. Pap smears were collected from each participant with a lubricated speculum ('Gel group') or a non-lubricated speculum ('No Gel group'). The primary outcome measures were the proportion of women with unsatisfactory cervical cytology smears and the mean numeric rating scale pain scores, while the secondary outcome

**Data Availability Statement:** All relevant data are within the paper and its Supporting Information files.

**Funding:** The author(s) received no specific funding for this work.

**Competing interests:** The authors have declared that no competing interests exist.

measures were the proportion of women who were willing to come for repeat testing and the cytological diagnosis of Pap-smear results.

## Results

The baseline socio-demographic variables were similar in both groups. There was no significant difference in the proportion of unsatisfactory cervical smear results between the two groups (13.6% vs. 21.2%, p = 0.359). However, the mean pain scores were significantly lower in the gel group than in the no gel group (45.04 vs. 87.96; p<0.001). An equal proportion of the participants in each group (90.9% vs. 90.9%; p > 0.999) were willing to come for repeat cervical smears in the future.

## Conclusion

Speculum lubrication did not affect the adequacy of cervical smears but significantly reduced the pain experienced during pap smear collection. Also, it did not significantly affect the willingness to come for repeat cervical smears in the future.

## Trial registration

The trial was registered with the **Pan-African Clinical Trial Registry** with a unique identification and registration number: **PACTR2020077533364675**.

## Introduction

Cancer of the cervix is the fourth most prevalent malignancy in females, with a projected incidence of 570,000 in 2018 [1]. It represents 6.6 percent of all gynaecological carcinomas and is responsible for 7.5 percent of all female malignancy mortality worldwide [1]. Out of the 311 000 cases of mortality estimated from carcinoma of the cervix annually, greater than 85% of these deaths happen in areas that are less developed [1].

Introducing screening for cancer of the cervix and improving its uptake, especially in developing countries, is key to decreasing the death burden from malignancy of the cervix. In addition, the usage of screening programs can be improved by increasing sensitization to the risk factors of carcinoma of the cervix, which include early coitarche, multiple sexual partners, HPV infections, extensive use of oral contraceptives, and HIV infections [2].

The conventional Papanicolaou test remains the most common screening method for cancer of the cervix [3]. In high-income countries, there are well-structured programs that enable the vaccination of girls against HPV and regular screening of women, resulting in a remarkable decrease in the development and complications of carcinoma of the cervix [4–6]. Screening helps to achieve early identification and intervention, which prevents up to 80% of cervical malignancies in high-income countries [4–6], whereas in poor-income countries, there is poor access to vaccination and testing, resulting in the diagnosis of cervical cancer, mainly in the advanced stages [7,8]. In addition, these countries lack access to the facilities required for the management of such advanced-stage diseases, giving rise to a greater death rate from malignancy of the cervix in poor countries [7,8].

As much as efforts are being made to improve vaccinations for HPV in order to prevent carcinoma of the cervix, early identification of premalignant lesions of the cervix through cervical Papanicolaou smear cytology screening remains a key factor in achieving a decline in the

development and complications of carcinoma of the cervix in low and middle-income countries where vaccination for Human Papilloma Virus services is limited [9].

The Papanicolaou test is a well-recognised, efficient, and reliable tool employed in the early identification of premalignant lesions of the cervix, resulting in a substantial decrease in the disease burden of carcinoma of the cervix [10]. It is cost-effective, and the technique is simple. Despite this, the uptake of Papanicolaou smears in our environment remains poor. Pain and discomfort associated with the examination of the vagina can discourage women from taking regular tests. Other factors that may hinder compliance include a lack of awareness, cost implications, anxiety, and cultural beliefs [7,11].

Insertion of a speculum for examination of the vagina is an important factor responsible for non-compliance with regular screening and repeat testing for carcinoma of the cervix because of the embarrassment, anxiety, pain, and discomfort associated with it [12]. In Australia, research seeking to find out the attitude of women concerning self-insertion compared to physician insertion of the speculum showed that 91% of the study population would prefer to insert the speculum by themselves rather than a physician doing it because of the embarrassment and discomfort associated with it [13].

During intercourse, lubrication is physiologically essential for easy penetration of the vagina, and the absence of optimal lubrication results in dyspareunia. So, we cannot justify inserting, without lubrication, a rigid instrument like a speculum into the vagina. Speculum lubrication should be employed to minimise discomfort and pain during vaginal examinations, thereby ultimately increasing compliance for screening. However, applying lubricating gel to the vagina is not encouraged by gynaecologic literature, and in addition, students and resident doctors in training are advised against lubricating the speculum while collecting samples due to the worry that it may interfere with the cytology results of cervical smears, often leading to inadequacy [10,14]. However, there is a paucity of convincing evidence to prove that using lubricating gel can prevent proper cytological analysis [15–17].

The aim of this study was to determine the effects of speculum lubrication on the adequacy of cervical cytology smears, determine if it decreases pain or discomfort in women undergoing cervical cancer screening by means of cervical smears, and also compare the proportion of women willing to come for repeat cervical smears in the future.

## Materials and methods

This work was a prospective randomised controlled trial of 132 women having cervical cancer screening at the gynaecologic clinic of Nnamdi Azikiwe University Teaching Hospital (NAUTH) in Nnewi, Anambra State, Nigeria, from August 21, 2020, to December 31, 2020. Ethical approval was obtained from the NAUTH Ethics Review Committee with the approval reference number: NAUTH/CS/66/VOL.12/098/2019/040.

The participants included pre-menopausal women of age not less than 25 years and postmenopausal women who required pap smear screening. Women excluded were virgins, recent sexual intercourse, pregnant women, women having their menstrual period, women with any overt cervical pathology and/or current treatment of any vaginal condition, women with vulvar pathologies, or those on hormone replacement therapy. Women with vaginitis, those undergoing vulvectomy or vaginectomy, and women who had fertility-sparing surgery were also excluded.

The sample size was based on the proportions of unsatisfactory cervical cytology smears, which is one of the primary outcome measure. The sample size was calculated using the formula for determining the minimum sample size for experimental studies (difference in proportions of unsatisfactory cervical cytology smears). Substituting for the values of the

proportion of unsatisfactory smears as found in a previous study [11].

$$n = \frac{[(Z_\beta + Z_{\alpha/2})^2 \text{ x } 2P(1 - P)]}{E^2}$$

where P (combined proportions of unsatisfactory smears in a previous study by Uygur et al [11]) = $(P_1 + P_2)$ / 2 = (0.01+ 0.005) / 2 = 0.0075, E = Effect size = $P_1$—$P_2$ = 0.01–0.005 = 0.005, $Z_\beta$ = corresponding Z value at 80% Power = 0.84, and $Z_{\alpha/2}$ = 1.96 (corresponding Z value at 95% confidence level). Adjusting for a finite population, N = 60 samples from the clinic in 3 months; $n_s = n^1$ / 1 + $n^1$/N, where $n_s$ = adjusted sample size, $n^1$ = calculated sample size, and N = population for study. Considering an attrition rate of 10.0%, the total number of subjects was 132 (66 in each arm).

The procedure was explained, and participants were counselled on the procedure. Pre-procedure precautions like abstinence from sexual intercourse for at least 24 hours prior to the procedure and avoidance of douching were observed. Trained research assistants (senior registrars) and the researchers were involved in the counselling of participants and the collection of the pap smears. Following written informed consent, selected women were randomly assigned to the gel group or no gel group. The sequence of randomization was generated by the computer via randomly permuted blocks (blocks of 4; allocation ratio: 1:1). To keep a similar number of subjects in each group, a block randomization method with blocks of 4 was used. An independent person who was not involved in the study performed the randomization using a computer software program available at http://mahmoodsaghaei.tripod.com/Softwares/randalloc.html. Allocation concealment was done using serially numbered opaque sealed envelopes, which contained either a piece of paper showing gel group (Group A), placing the participant into the group using speculum lubrication, or a piece of paper displaying no gel group (Group B), placing the participant into the group receiving dry speculum (without speculum lubrication). The envelopes were kept and opened by an independent person. Once the envelope was opened, the allocation of participants was not changed. The histopathologist and participants were blinded, so they were not aware of the intervention arm; hence, they were double blind. The primary outcome of the study was proportions of unsatisfactory cervical cytology smears, an outcome that was determined by the histopathologists as one of the outcome assessor. We can state that except for the researcher, the histopathologist and participants were blinded and so were not aware of the intervention arm, thereby preventing bias and giving more credence to the study. Regarding the participants, the application of the lubricants to the speculum was made when the participants were already in lithotomy or dorsal position, with the faces not facing the researchers when collecting the smears. Also, the speculum preparation was made in the other section of the examining room to ensure the blinding of the intervention assignment.

The study procedure required the participants to be placed in dorsal position with the legs flexed at the knee and adducted at the hips. Following identification of the appropriate size speculum, 2 ml of KY jelly (Dionel, Maryland, USA) was applied to the entire external surfaces of the metal Cusco's speculum and inserted gently into the vagina to visualise the cervix. A non-lubricated speculum was used for the control group. A dry gauze swab was used to clear any mucous on the cervix. A cytobrush (Liqui-PREP[TM] LGM International, Inc., Melbourne, USA) was used to take the sample for the pap smear. Having obtained the sample, the head of the brush was removed and put into the liquid preparative collection vial containing a liquid-based medium. (Liqui-PREP[TM] LGM International, Inc., Melbourne, USA) The specimens in the vial were then carefully mixed with the liquid-based medium. Then, the speculum was gently withdrawn. The patient was cleaned and counselled on the outcome. After each smear,

the patient was asked to rate her pain at the end of the procedure using a validated numerical rating scale for pain from 0 (no discomfort) to 10 (most discomfort) to indicate if she was willing to come for repeat testing in the future. The numeric rating scale for pain was used because of its authenticity and universal acceptance for the evaluation of pain [11].

In the histopathology laboratory of NAUTH, the labelled vial was transferred into a centrifuge tube with the same quantity of cleaning solution. The fluid was then centrifuged for 10 minutes at 1400 rpm, and the supernatant was removed. Smears were made from the sediments after mixing with the cellular base solution (Liqui-PREP$^{TM}$ LGM International, Inc., Melbourne, USA), stained by the Papanicolaou staining method, and reported using the Bethesda system, stating whether the sample was satisfactory or not and the reason for unsatisfactory results. Samples were recorded as "unsatisfactory" if 75% of the cells were obscured by blood or inflammation, or if drying artefacts or gel overlays were present. Otherwise, they were recorded as "satisfactory" [10,18]. The patients with unsatisfactory smear results and positive cytological results were referred to the managing gynaecological unit for management and follow-up, while those with negative cytological results were counselled for repeat testing.

The primary outcome measures were the proportion of women with unsatisfactory cervical cytology smears and the mean numeric rating scale pain scores. While the secondary outcome measures were the proportion of women who were willing to come for repeat testing and the cytological diagnosis of pap smear results.

Statistical Packages for Social Sciences (SPSS), IBM Corp. version 25.0, was employed in the analysis of the results. Tables were used to represent the collected data. Means and standard deviations were used to represent continuous data. The Mann-Whitney U test was used to assess non-parametric variables, and the chi-squared test was used for categorical variables. A two-sided p value of <0.05 is considered statistically significant. A subgroup analysis of the association between categories of pain and parity was done.

## Results

One hundred and forty-nine (149) women were assessed for eligibility; however, 132 participants met the inclusion criteria and were randomised into the gel group (n = 66) and the no gel group (n = 66). A flow diagram describing how the participants flowed through the study is shown in Fig 1.

Table 1 shows the socio-demographic characteristics of participants in both groups. In terms of educational qualification, 89 (67.4%) of the participants had tertiary education, while 28% and 4.5% attained secondary and primary education, respectively. There was no significant difference in the baseline sociodemographic data between the two groups.

There was no significant difference in the proportion of unsatisfactory cytology smears (13.6% vs. 21.2%; p = 0.359) and normal Pap smear results (100.0% vs. 98.1%; p = 0.477) in the gel group and no gel group, respectively. The reasons for the unsatisfactory smear results were obscurity by marked inflammation (11.1% vs. 14.3%; p = 0.691) and the presence of scanty cells (88.9% vs. 85.7%; p = 0.691) in the Gel and No Gel groups, respectively. Drying artefacts due to gel were not reported by the histopathologist. This is shown in Table 2.

The mean pain scores were statistically lower in the gel group than in the no gel group (45.04 vs. 87.96; p<0.001). While 3 people from the no gel group had scores of 9 and 10, the pain scores in the gel group were less than 9. Additionally, 69.7%, 22.7%, and 7.6% of the women in the gel group had mild, moderate, and severe pain, respectively, while 3.0%, 68.2%, and 18.2% in the no gel group had mild, moderate, and severe pain, respectively (p<0.001). This is shown in Table 3.

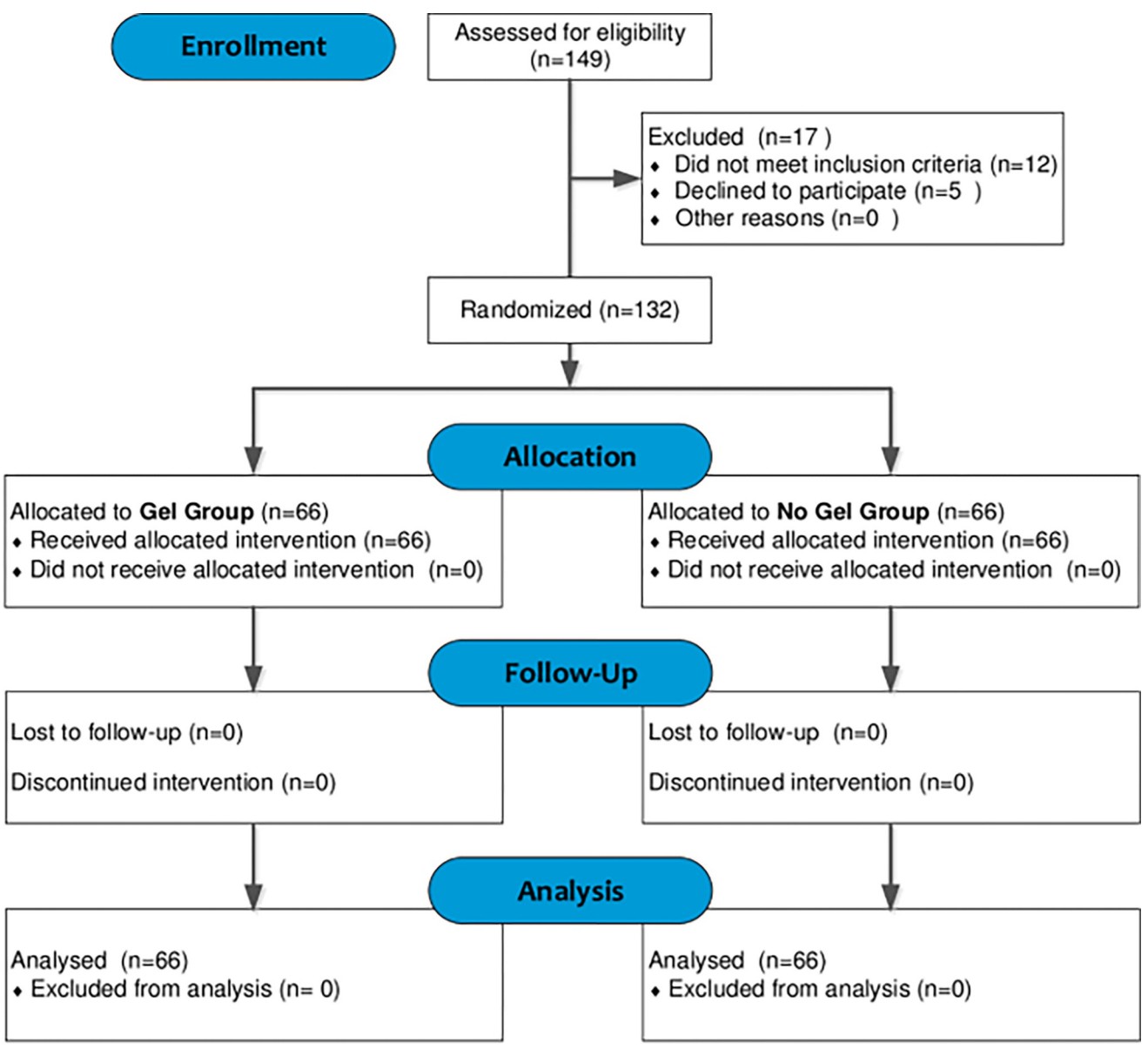

**Fig 1. Consort flowchart for study participants.**

The association between the categories of pain and parity among the participants is shown in Table 4. The pain scores appear to be higher at the extremes of parity.

The majority of the participants in both groups (90.9% vs. 90.9%; p > 0.999) were willing to come for repeat cervical smears in the future after undergoing initial cervical cancer screening. Although more participants in the gel group (98.5% vs. 92.4%; p = 0.208) were more satisfied with the smear collection procedure than in the no gel group, the difference was not statistically significant. Some participants were unwilling to repeat the Pap smear test because the procedure was discomforting (33.3% vs. 16.7%; p = 0.691), painful (66.7% vs. 66.7%; p = 0.691), and very uncomfortable (0.0% vs. 16.7%; p = 0.691) in the gel and no gel groups, respectively. This is shown in Table 5.

**Table 1. Socio-demographic characteristics of the participants.**

| Variables | | Study Group | | |
|---|---|---|---|---|
| | | GEL (%) | No GEL(%) | Total (%) |
| **Age** | 25–34 | 25 (37.9) | 23 (34.8) | 48 (36.4) |
| **(in years)** | 35–44 | 19 (28.8) | 23 (34.8) | 42 (31.8) |
| | 45–54 | 17 (25.8) | 15 (22.7) | 32 (24.2) |
| | 55 and above | 5 (7.6) | 5 (7.6) | 10 (7.6) |
| **Marital status** | Single | 16 (24.2) | 16 (24.2) | 32 (24.2) |
| | Married | 43 (65.2) | 47 (71.2) | 90 (68.2) |
| | Separated/Divorced | 1 (1.5) | 0 (0.0) | 1 (0.8) |
| | Widowed | 6 (9.1) | 3 (4.5) | 9 (6.8) |
| **Tribe** | Igbo | 65 (98.5) | 65 (98.5) | 130 (98.5) |
| | Ijaw | 1 (1.5) | 0 (0.0) | 1 (0.8) |
| | Benin | 0 (0.0) | 1 (1.5) | 1 (0.8) |
| **Educational status** | Primary | 4 (6.1) | 2 (3.0) | 6 (4.5) |
| | Secondary | 18 (27.3) | 19 (28.8) | 37 (28.0) |
| | Tertiary | 44 (66.7) | 45 (68.2) | 89 (67.4) |
| **Parity** | $P_0$ | 30 (45.5) | 31 (47.0) | 61 (46.1) |
| | $P_{1-2}$ | 13 (19.7) | 9 (13.7) | 22 (16.7) |
| | $P_{\geq 3}$ | 23 (34.8) | 26 (39.4) | 49 (37.2) |

## Discussion

The practice of speculum lubrication for vaginal examination is varied due to the fear that lubricants can alter the adequacy of cytological smears. However, recent studies documenting the relationship between speculum lubrication and Pap smear accuracy have shown evidence supporting that speculum lubrication does not disrupt the evaluation of cytological smears [19,20].

Our findings revealed that there was no significant difference in the proportion of unsatisfactory cytology smear results in the gel group or no gel group, respectively. This was similar to the findings in previous studies [10,11,18,21]. However, this was in contradiction to the findings of Charoenkwan et al. and Kosus et al., which showed statistically significant greater numbers of unsatisfactory smears in the participants that received speculum lubrication [22,23]. The reason for the differing report by Charoenkwan et al. may be because there was

**Table 2. Comparison of cytology smear adequacy and cytological diagnosis of Pap Smears results among the study participants and reasons for unsatisfactory smear results as reported by the histopathologist.**

| Variables | | Study Group | | | p-value |
|---|---|---|---|---|---|
| | | GEL (%) | No GEL (%) | Total (%) | |
| **Smear sample status** | Satisfactory | 57 (86.4) | 52 (78.8) | 109 (82.6) | 0.359 |
| | Unsatisfactory | 9 (13.6) | 14 (21.2) | 23 (17.4) | |
| **PAP smear result** | LSIL | 0 (0.0) | 1 (1.9) | 1 (0.9) | 0.477† |
| | NILM | 57 (100.0) | 51 (98.1) | 108 (99.1) | |
| **Reason for unsatisfactory result (total unsatisfactory results = 23)** | Obscured by marked inflammation | 1 (11.1) | 2 (14.3) | 3 (13.0) | 0.691† |
| | Scanty cells* | 8 (88.9) | 12 (85.7) | 20 (87.0) | |
| | Total | 9 (100.0) | 14 (100.0) | 23 (100.0) | |

LSIL = Low Grade Squamous Intraepithelial Lesion; NILM = Negative for Intraepithelial lesion or malignancy.

* Drying artifact due to gel was not reported by the histopathologist.

**Table 3. Comparison of Pain scores of the participants.**

| Variables | | Study Group | | Total (%) | p-value |
|---|---|---|---|---|---|
| | | GEL (%) | No GEL (%) | | |
| Numeric rating scale for pain (0 = no pain, 10 = worst pain) | 1 | 23 (34.8) | 1 (1.5) | 24 (18.2) | |
| | 2 | 13 (19.7) | 1 (1.5) | 14 (10.6) | |
| | 3 | 10 (15.2) | 0 (0.0) | 10 (7.6) | |
| | 4 | 2 (3.0) | 12 (18.2) | 14 (10.6) | |
| | 5 | 6 (9.1) | 17 (25.8) | 23 (17.4) | |
| | 6 | 7 (10.6) | 16 (24.2) | 23 (17.4) | |
| | 7 | 2 (3.0) | 11 (16.7) | 13 (9.8) | |
| | 8 | 3 (4.5) | 5 (7.6) | 8 (6.1) | |
| | 9 | 0 (0.0) | 1 (1.5) | 1 (0.8) | |
| | 10 | 0 (0.0) | 2 (3.0) | 2 (1.5) | |
| Chi-square association of pain Categories and study groups | Mild | 46 (69.7) | 2 (3.0) | 48 (36.4) | <0.001 |
| | Moderate | 15 (22.7) | 45 (68.2) | 60 (45.5) | |
| | Severe | 5 (7.6) | 19 (28.8) | 24 (18.2) | |
| **Mann-Whitney U Comparison of pain** | Mean Rank pain score | 45.04 | 87.96 | | <0.001* |

Mild = 1–3, moderate = 4–6, severe = 7–10. There is significant higher pain in the No-Gel group compared to the Gel group (p < 0.001).

direct contamination of the smears with gel in their study as opposed to the speculum lubrication employed in our present study. Similarly, in Kosus et al., the risk of significantly higher inadequate results in the gel group could be because of the personnel that collected the smears, as the majority were inexperienced trainees [23]. The experience of the provider has been implicated as one of the risk factors for smear inadequacy [11,18,21]. Also, there are some reports in the literature that suggest that lubricants can affect the results of cervical cytology [24,25]. In most of these studies, lubricant gel was added intentionally to liquid-based cytology specimens. Most of these study designs do not reflect actual clinical practice.

The Bethesda system requires that Papanicolaou preparations include enough cells to cover 10% of a slide [18]. If 75% of the epithelial cells are obscured by blood, inflammation, or artefacts, the slide is considered unsatisfactory [18,21]. The reasons for the unsatisfactory smears in this study were scanty cells (87%) and obscurity by inflammation (13%). This finding was similar between the two groups. Gel overlay or obscurity by gel was not recorded as a reason

**Table 4. Association between category of pain and parity among the participants.**

| Parity | Pain Category | Study Arms (%) | | p-value |
|---|---|---|---|---|
| | | GEL | NO GEL | |
| $P_0$ | Mild | 21 (70.0) | 2 (6.5) | **<0.001** |
| | Moderate | 6 (20.0) | 19 (61.3) | |
| | Severe | 3 (10.0) | 10 (32.3) | |
| $P_{1-2}$ | Mild | 8 (61.5) | 0 (0.0) | 0.013 |
| | Moderate | 4 (30.8) | 7 (77.8) | |
| | Severe | 1 (7.7) | 2 (22.2) | |
| $P_{\geq 3}$ | Mild | 17 (73.9) | 0 (0.0) | <0.01 |
| | Moderate | 5 (21.7) | 19 (73.1) | |
| | Severe | 1 (4.3) | 7 (26.9) | |

Mild = 1–3, moderate = 4–6, severe = 7–10.

**Table 5. Satisfaction for Pap smear procedure and willingness for repeat of Pap smear among the study participants and their reasons for unwillingness to repeat Pap smear test.**

| Variables | | Study Group | | | p-value |
|---|---|---|---|---|---|
| | | GEL (%) | No GEL(%) | Total (%) | |
| **Satisfied with the PAP smear procedure?** | YES | 65 (98.5) | 61 (92.4) | 126 (95.5) | 0.208† |
| | NO | 1 (1.5) | 5 (7.6) | 6 (4.5) | |
| | Total | 66 (100) | 66 (100) | 132 (100) | |
| **Are you willing to come for repeat testing in future?** | YES | 60 (90.9) | 60 (90.9) | 120 (90.9) | >0.999 |
| | NO | 6 (9.1) | 6 (9.1) | 12 (9.1) | |
| | Total | 66 (100) | 66 (100) | 132 (100) | |
| **Reason for unwillingness** | Discomforting | 2 (33.3) | 1 (16.7) | 3 (25.0) | >0.999† |
| | Painful | 4 (66.7) | 4 (66.7) | 8 (66.7) | |
| | Very uncomfortable | 0 (0.0) | 1 (16.7) | 1 (8.3) | |
| | Total | 6 (100.0) | 6 (100.0) | 12 (100.0) | |

for unsatisfactory results. This is similar to the findings by Amies et al. and Gilson et al., where the reason for the unsatisfactory result was obscurity by blood, inflammation, and gel overlay, which was not documented as a reason for unsatisfaction [10,18].

This study has revealed that higher pain scores were recorded in the no-gel group compared to the gel group (p<0.001). This finding agrees with previous studies by Uygur et al. and Gungorduk et al., where the pain scores were significantly higher in the arm that did not use speculum lubrication [11,26]. However, our findings differ from the work of Gilson et al., who reported that speculum lubrication did not affect pain or discomfort [10]. The difference between the present study and that of Gilson et al. may be because of the method of assessment of the pain used. In the study by Gilson et al., pain was assessed using the Wong-Baker Faces Pain Rating Scale, which is a less effective method of pain measurement. In this study, a numerical rating scale for pain, which has been shown to be widely appropriate for pain evaluation globally, was utilised.

All the samples came out negative for intraepithelial lesions or malignancies, except one that was a low-grade squamous intraepithelial lesion. There was no diagnosis of invasive cancer or a high-grade squamous intraepithelial lesion. This absence or insignificant number of positive cytological reports of epithelial lesions has also been reported in other studies [21,24]. This might be explained by the fact that the study population was more of the asymptomatic younger age group, healthy volunteers, and not the age group with a higher risk of positive cytological reports of epithelial lesions.

When the pain scores assessed were compared according to the parity of the participants, it was observed that the severity of the pain was seen at the extremes of parity. The reason for this peculiar finding may be that in the nulliparous state, the vagina and introitus may be narrower than in the parous state. However, the pain may also be more severe at higher parity because women with higher parity may have completed family size, are older, or may be postmenopausal, resulting in vaginal atrophy and vaginitis. These conditions are expected to elicit more pain reactions among the participants.

This study revealed that more participants in the gel group (98.5% vs. 92.4%; p = 0.208) were more satisfied with the pap smear collection procedure than in the no gel group, but the difference was not statistically significant. In addition, an equal proportion of the participants in each group (90.9% vs. 90.9%; p > 0.999) were willing to come for repeat cervical smears in the future after undergoing initial cervical cancer screening. This finding is not only interesting but also encouraging. This could be explained by the fact that more than half (67.4%) of

the study population had tertiary education and 28% had secondary education, resulting in a high level of awareness, knowledge, and need for cervical screening among the study population. In a previous review by Chorley et al., it was revealed that a single negative experience prevented some women from re-attending screenings, even if they had multiple positive previous experiences to draw upon. However, the studies involved in the Chorley et al. systematic review were not randomised controlled trials and did not compare the failure at re-screening in women that received speculum lubrication versus no speculum lubrication [27].

It was also noted that the reasons why some participants were unwilling to come for repeat testing in the future were due to pain (66.7%) and discomfort (25%). Although these findings do not differ among the two groups, it still helps to buttress the fact that pain while undergoing pap smear collection is a strong reason why some women would not present for repeat testing.

The strength of the study was that it was a double-blind, randomised, controlled study. The study also employed liquid-based cytology, thereby forming a literature base for future comparison, as most of the studies in the literature employed the conventional Pap smear technique. However, the limitation was that only one type of water-based lubricant (KY-Jelly) was used, so results may not be applied to other kinds of lubricants. Also, our study only utilised metal speculums, so the findings may not be applied to plastic types of speculum. Perception of pain is complex and multifactorial, and so cultural, genetic, and environmental factors may affect these results in different populations. The study was single-centre-based; hence, multi-centre, similar randomised studies are needed. Also, we did not document the menopausal status of the participants.

## Conclusion

In conclusion, speculum lubrication did not affect the adequacy of the cervical cytology smears, satisfaction with the pap smear procedure, or willingness to come for repeat cervical smears in the future after initial cervical cancer screening, but significantly reduced the pain and discomfort experienced at pap smear collection.

## CONSORT statement

The study adhered to CONSORT guidelines [28].

## Supporting information

**S1 Checklist. CONSORT 2010 checklist of information to include when reporting a randomised trial\*.**
(DOCX)

**S1 File.**
(XLSX)

**S2 File.**
(DOCX)

## Acknowledgments

The authors sincerely appreciate everyone who contributed to the success of this study, especially the patients that were recruited into the study and the NAUTH, Nnewi, Nigeria.

## Author Contributions

**Conceptualization:** Chito P. Ilika.

**Data curation:** Chito P. Ilika.

**Formal analysis:** Michael E. Chiemeka, Frances N. Ilika, Valentine C. Ilika, Chidinma C. Okafor, Onyeka C. Ekwebene, Obinna K. Nnabuchi.

**Methodology:** Chito P. Ilika, George U. Eleje, Michael E. Chiemeka, Frances N. Ilika, Joseph I. Ikechebelu, Valentine C. Ilika, Emmanuel O. Ugwu, Ifeanyichukwu J. Ofor, Onyecherelam M. Ogelle, Osita S. Umeononihu, Johnbosco E. Mamah, Chinedu L. Olisa, Chijioke O. Ezeigwe, Malarchy E. Nwankwo, Chukwuemeka J. Ofojebe, Chidinma C. Okafor, Onyeka C. Ekwebene, Obinna K. Nnabuchi, Chigozie G. Okafor.

**Supervision:** George U. Eleje, Joseph I. Ikechebelu.

**Writing – original draft:** Chito P. Ilika, George U. Eleje, Michael E. Chiemeka, Frances N. Ilika, Joseph I. Ikechebelu, Valentine C. Ilika, Emmanuel O. Ugwu, Ifeanyichukwu J. Ofor, Onyecherelam M. Ogelle, Osita S. Umeononihu, Johnbosco E. Mamah, Chinedu L. Olisa, Chijioke O. Ezeigwe, Malarchy E. Nwankwo, Chukwuemeka J. Ofojebe, Chidinma C. Okafor, Onyeka C. Ekwebene, Obinna K. Nnabuchi, Chigozie G. Okafor.

**Writing – review & editing:** Chito P. Ilika, George U. Eleje, Michael E. Chiemeka, Frances N. Ilika, Joseph I. Ikechebelu, Valentine C. Ilika, Emmanuel O. Ugwu, Ifeanyichukwu J. Ofor, Onyecherelam M. Ogelle, Osita S. Umeononihu, Johnbosco E. Mamah, Chinedu L. Olisa, Chijioke O. Ezeigwe, Malarchy E. Nwankwo, Chukwuemeka J. Ofojebe, Chidinma C. Okafor, Onyeka C. Ekwebene, Obinna K. Nnabuchi, Chigozie G. Okafor.

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
