## [Decision Letter · Decision Letter 0]

30 Jan 2024

PONE-D-23-28708Effects of speculum lubrication on cervical smears for cervical cancer screening: a double blind randomized controlled trialPLOS ONE

Dear Dr. OKAFOR,

Thank you for submitting your manuscript to PLOS ONE. After careful consideration, we feel that it has merit but does not fully meet PLOS ONE’s publication criteria as it currently stands. Therefore, we invite you to submit a revised version of the manuscript that addresses the points raised during the review process.

We look forward to receiving your revised manuscript.

Kind regards,

David Chibuike Ikwuka, Ph.D.

Academic Editor

PLOS ONE

Journal Requirements:

2. In the online submission form, you indicated that "Data is available upon request from the authors" 

4. Please remove your figures from within your manuscript file, leaving only the individual TIFF/EPS image files, uploaded separately. These will be automatically included in the reviewers’ PDF.

Additional Editor Comments:

**C**arryout the revisions and react to the comments of the reviewers if need be. 

Reviewers' comments:

Reviewer's Responses to Questions

**Comments to the Author**

1. Is the manuscript technically sound, and do the data support the conclusions?

Reviewer #1: Yes

Reviewer #2: Yes

2. Has the statistical analysis been performed appropriately and rigorously? 

Reviewer #1: Yes

Reviewer #2: Yes

3. Have the authors made all data underlying the findings in their manuscript fully available?

Reviewer #1: No

Reviewer #2: No

4. Is the manuscript presented in an intelligible fashion and written in standard English?

Reviewer #1: No

Reviewer #2: Yes

5. Review Comments to the Author

Reviewer #1: 1. There are many typos and grammatical mistakes throughout the text.

2. Please describe the sample size calculation using more words. What outcome is the sample size based on?

3. In some places the arms are described using gel and no gel or lubrication and no lubrication. Please be consistent.

4. In the statistics section - "Man Whitney U test was used to assess non-parametric variables" -- what does this mean?

5. Indicate whether two-sided p values were used.

6. In the results you indicate that there were no significant differences between the groups. Testing the sociodemographic characteristics was not described in the stats section. In fact, you should not do this. So please remove this sentence, and remove the chi-square value and p value from Table 1.

7. The chi-square value can be removed (but keep the p values) from tables 2, 3, 4, 6, 7

8. The test value can be removed from table 8.

9. Remove the sum of ranks, ManWhitney U, and Z avlue from table 5.

9. One subgroup was investigated - this was parity. Was this pre-specified?

10. For parity - there are too many small groups. Pleas consider collapsing some of the groups e.g. 0, 1-2, 3+

11. there are many small tables - can some of these be combined?

12. How was pain assessed? Is this a validated scale?

13. The reason for increased pain in the high parity group is speculated as being due to menopause. Was menopausal status of the women collected? Can you adjust the parity association for menopausal status, or for age? What happens?

14. The trial is described as double blind. The participants surely knew which arm they were allocated to? Similarly, the person performing the pap smear knew which arm the women were allocated to. Please discuss blinding more thoroughly in the manuscript.

Reviewer #2: Introduction.

The prevalence quoted in your manuscript is not in tandem with information in the cited reference and therefore not correct. Ca cervix is the 3rd most common female cancer globally and the 2nd in Africa as contained in your referenced publication.

Methodology

Line 106; Change statement to ... because its authenticity .....

Results

Table 5 is not clear and appears confusing.

Table 6

Since the findings have already established that NO GEL group felt more pain than the GEL group, the comparison here should have been among the various parities while inserting the speculum whether lubricated or not. This is also captured in your discussion. Comparing the level of pain at different parities with GEL on NO GEL group is not necessary and appears confusing.

Discussion

Reference should be cited after the claim in line 177

6. PLOS authors have the option to publish the peer review history of their article (what does this mean?). If published, this will include your full peer review and any attached files.

Reviewer #1: No

Reviewer #2: No

---

## [Author Response · Author response to Decision Letter 0]

4 Apr 2024

04/04/2024

From 

Corresponding author (Dr. Chigozie Geoffrey Okafor)

To 

Editor

PLOS ONE

Dear Editor,

Re: submission of the response to reviewers’ comments on manuscript id PONE-D-23-28708 entitled ‘Effects of speculum lubrication on cervical smears for cervical cancer screening: a double blind randomized clinical trial’. Thank you for your thorough review of the above-named manuscript. I have enclosed a point-by-point response to the queries/comments raised by the Editor and the reviewers. The authors are hopeful that our corrections/revisions will be acceptable to you and the reviewers.

REVIEWER 1 COMMENTS TO THE AUTHORS

1. There are many typos and grammatical mistakes throughout the text.

Authors’ Responses

Thorough language editing done.

2. Please describe the sample size calculation using more words. What outcome is the sample size based on?

Authors’ Response

The sample size was based on the proportions of unsatisfactory cervical cytology smears, which is one of the primary outcome measure. The sample size was calculated using the formula for determining the minimum sample size for experimental studies (difference in proportions of unsatisfactory cervical cytology smears). Substituting for the values of the proportion of unsatisfactory smears as found in a previous study; 11

n = [(Zβ + Zα/2)2 x 2P(1−P)]

 E2

where P (combined proportions of unsatisfactory smears in a previous study by Uygur et al11) = (P1 + P2) / 2 = (0.01+ 0.005) / 2 = 0.0075, E = Effect size = P1 - P2 = 0.01- 0.005 = 0.005, Zβ = corresponding Z value at 80% Power = 0.84, and Zα/2 = 1.96 (corresponding Z value at 95% confidence level). Adjusting for a finite population, N = 60 samples from the clinic in 3 months; ns = n1 / 1 + n1/N, where ns = adjusted sample size, n1 = calculated sample size, and N = population for study. Considering an attrition rate of 10.0%, the total number of subjects was 132 (66 in each arm). The second primary outcome measure of the mean numeric rating scale pain scores values for the calculated sample size was much lower compared to the sample size using the proportions of unsatisfactory cervical cytology smears. Therefore, the sample size using the proportions of unsatisfactory cervical cytology smears was utilized in the study. See page 6,

3. In some places the arms are described using gel and no gel or lubrication and no lubrication. Please be consistent.

Authors’ Response

Thank you for pointing this out. Corrections made. Lubrication and no lubrication replaced with Gel and No gel. For uniformity, gel and no gel group now used throughout the manuscript.

4. In the statistics section - "Man Whitney U test was used to assess non-parametric variables" -- what does this mean?

Authors Response’

Mann Whitney U test was used in the statistical analysis because the test variable (pain scale) was not normally distributed.

5. Indicate whether two-sided p values were used

Authors’ Response’

Yes, the two sided –values were used. We append in the manuscript that: A two-sided p value <0.05 is considered statistically significant. See line 135-136 on page 8. 

6. In the results you indicate that there were no significant differences between the groups. Testing the sociodemographic characteristics was not described in the stats section. In fact, you should not do this. So please remove this sentence, and remove the chi-square value and p value from Table 1.

Authors Response’ 

Thank you for drawing our attention to it. This has been deleted as shown below.

Table 1: Socio-demographic characteristics of the participants

Variables Study Group 

 GEL (%) No GEL(%) Total (%)

Age 25-34 25 (37.9) 23 (34.8) 48 (36.4)

(in years) 35-44 19 (28.8) 23 (34.8) 42 (31.8)

 45-54 17 (25.8) 15 (22.7) 32 (24.2)

 55 and above 5 (7.6) 5 (7.6) 10 (7.6)

Marital status Single 16 (24.2) 16 (24.2) 32 (24.2)

 Married 43 (65.2) 47 (71.2) 90 (68.2)

 Separated/Divorced 1 (1.5) 0 (0.0) 1 (0.8)

 Widowed 6 (9.1) 3 (4.5) 9 (6.8)

Tribe Igbo 65 (98.5) 65 (98.5) 130 (98.5)

 Ijaw 1 (1.5) 0 (0.0) 1 (0.8)

 Benin 0 (0.0) 1 (1.5) 1 (0.8)

Educational status Primary 4 (6.1) 2 (3.0) 6 (4.5)

 Secondary 18 (27.3) 19 (28.8) 37 (28.0)

 Tertiary 44 (66.7) 45 (68.2) 89 (67.4)

Parity P0 30 (45.5) 31 (47.0) 61 (46.1)

 P1-2 13 (19.7) 9 (13.7) 22 (16.7)

 P≥3 23 (34.8) 26 (39.4) 49 (37.2)

7. The chi-square value can be removed (but keep the p values) from tables 2, 3, 4, 6, 7

Authors’ Response’

Done. See the new table 2, 3 4, and 5 as shown below.

Table 2: Comparison of cytology smear adequacy and cytological diagnosis of Pap Smears results among the study participants and reasons for unsatisfactory smear results as reported by the histopathologist

Variables Study Group p-value

 GEL (%) No GEL (%) Total (%) 

Smear sample status Satisfactory 57 (86.4) 52 (78.8) 109 (82.6) 0.359

 Unsatisfactory 9 (13.6) 14 (21.2) 23 (17.4) 

PAP smear result LSIL 0 (0.0) 1 (1.9) 1 (0.9) 0.477†

 NILM 57 (100.0) 51 (98.1) 108 (99.1)

Reason for unsatisfactory result (total unsatisfactory results = 23) Obscured by marked inflammation 1 (11.1) 2 (14.3) 3 (13.0) 0.691†

 Scanty cells* 8 (88.9) 12 (85.7) 20 (87.0) 

 Total 9 (100.0) 14 (100.0) 23 (100.0)

LSIL = Low Grade Squamous Intraepithelial Lesion; NILM= Negative for Intraepithelial lesion or malignancy

* Drying artifact due to gel was not reported by the histopathologist

Table 3: Comparison of Pain scores of the participants 

Variables Study Group Total (%) p-value

 GEL (%) No GEL (%) 

Numeric rating scale for pain 

(0 = no pain, 10 = worst pain) 1 23 (34.8) 1 (1.5) 24 (18.2) 

 2 13 (19.7) 1 (1.5) 14 (10.6) 

 3 10 (15.2) 0 (0.0) 10 (7.6) 

 4 2 (3.0) 12 (18.2) 14 (10.6) 

 5 6 (9.1) 17 (25.8) 23 (17.4) 

 6 7 (10.6) 16 (24.2) 23 (17.4) 

 7 2 (3.0) 11 (16.7) 13 (9.8) 

 8 3 (4.5) 5 (7.6) 8 (6.1) 

 9 0 (0.0) 1 (1.5) 1 (0.8) 

 10 0 (0.0) 2 (3.0) 2 (1.5) 

Chi-square association of

pain Categories and study groups Mild 46 (69.7) 2 (3.0) 48 (36.4) <0.001

 Moderate 15 (22.7) 45 (68.2) 60 (45.5) 

 Severe 5 (7.6) 19 (28.8) 24 (18.2) 

Mann-Whitney U Comparison of pain Mean Rank pain score 45.04 87.96 <0.001*

Mild= 1-3, moderate = 4-6, severe = 7-10. There is significant higher pain in the No-Gel group compared to the Gel group (p < 0.001)

Table 4: Association between category of pain and parity among the participants

Parity Pain Category Study Arms (%) p-value

 GEL NO GEL 

P0 Mild 21 (70.0) 2 (6.5) <0.001

 Moderate 6 (20.0) 19 (61.3) 

 Severe 3 (10.0) 10 (32.3) 

P1-2 Mild 8 (61.5) 0 (0.0) 0.013

 Moderate 4 (30.8) 7 (77.8) 

 Severe 1 (7.7) 2 (22.2) 

P ≥3 Mild 17 (73.9) 0 (0.0) <0.01

 Moderate 5 (21.7) 19 (73.1) 

 Severe 1 (4.3) 7 (26.9) 

Mild= 1-3, moderate = 4-6, severe = 7-10.

Table 5: Satisfaction for Pap smear procedure and willingness for repeat of Pap smear among the study participants and their reasons for unwillingness to repeat Pap smear test

Variables Study Group p-value

 GEL (%) No GEL(%) Total (%) 

Satisfied with the PAP smear procedure? YES 65 (98.5) 61 (92.4) 126 (95.5) 0.208†

 NO 1 (1.5) 5 (7.6) 6 (4.5) 

 Total 66 (100) 66 (100) 132 (100) 

Are you willing to come for repeat testing in future? YES 60 (90.9) 60 (90.9) 120 (90.9) >0.999

 NO 6 (9.1) 6 (9.1) 12 (9.1) 

 Total 66 (100) 66 (100) 132 (100) 

Reason for unwillingness Discomforting 2 (33.3) 1 (16.7) 3 (25.0) >0.999†

 Painful 4 (66.7) 4 (66.7) 8 (66.7) 

 Very uncomfortable 0 (0.0) 1 (16.7) 1 (8.3) 

 Total 6 (100.0) 6 (100.0) 12 (100.0)

8. The test value can be removed from table 8.

Authors’ Response

Done. See table 5 (table 7 and 8 now merged to become new table 5) as shown below.

Table 5: Satisfaction for Pap smear procedure and willingness for repeat of Pap smear among the study participants and their reasons for unwillingness to repeat Pap smear test

Variables Study Group p-value

 GEL (%) No GEL(%) Total (%) 

Satisfied with the PAP smear procedure? YES 65 (98.5) 61 (92.4) 126 (95.5) 0.208†

 NO 1 (1.5) 5 (7.6) 6 (4.5) 

 Total 66 (100) 66 (100) 132 (100) 

Are you willing to come for repeat testing in future? YES 60 (90.9) 60 (90.9) 120 (90.9) >0.999

 NO 6 (9.1) 6 (9.1) 12 (9.1) 

 Total 66 (100) 66 (100) 132 (100) 

Reason for unwillingness Discomforting 2 (33.3) 1 (16.7) 3 (25.0) >0.999†

 Painful 4 (66.7) 4 (66.7) 8 (66.7) 

 Very uncomfortable 0 (0.0) 1 (16.7) 1 (8.3) 

 Total 6 (100.0) 6 (100.0) 12 (100.0)

9. Remove the sum of ranks, ManWhitney U, and Z values from table 5.

Authors’ Response

Done. See the new table 3 (table 4 &5 merged to become new table 3) as shown below.

Table 3: Comparison of Pain scores of the participants 

Variables Study Group Total (%) p-value

 GEL (%) No GEL (%) 

Numeric rating scale for pain 

(0 = no pain, 10 = worst pain) 1 23 (34.8) 1 (1.5) 24 (18.2) 

 2 13 (19.7) 1 (1.5) 14 (10.6) 

 3 10 (15.2) 0 (0.0) 10 (7.6) 

 4 2 (3.0) 12 (18.2) 14 (10.6) 

 5 6 (9.1) 17 (25.8) 23 (17.4) 

 6 7 (10.6) 16 (24.2) 23 (17.4) 

 7 2 (3.0) 11 (16.7) 13 (9.8) 

 8 3 (4.5) 5 (7.6) 8 (6.1) 

 9 0 (0.0) 1 (1.5) 1 (0.8) 

 10 0 (0.0) 2 (3.0) 2 (1.5) 

Chi-square association of

pain Categories and study groups Mild 46 (69.7) 2 (3.0) 48 (36.4) <0.001

 Moderate 15 (22.7) 45 (68.2) 60 (45.5) 

 Severe 5 (7.6) 19 (28.8) 24 (18.2) 

Mann-Whitney U Comparison of pain Mean Rank pain score 45.04 87.96 <0.001*

Mild= 1-3, moderate = 4-6, severe = 7-10. There is significant higher pain in the No-Gel group compared to the Gel group (p < 0.001)

10. One subgroup was investigated. This was parity. Was it pre-specified?

Authors’ Response

Your observation is quite valid. We wanted to see if parity has an effect on pain perception among the participants. This has been included in the statistical analysis statement as shown below.

Statistical Packages for Social Sciences (SPSS), IBM Corp. version 25.0, was employed in the analysis of the results. Tables were used to represent the collected data. Means and standard deviations were used to represent continuous data. Mann-Whitney U test was used to assess non-parametric variables, and the chi squared test was used for categorical variables. Statistical significance was deduced at a p-value of less than 0.05. A subgroup analysis of the association between categories of pain and parity was done. See line 136 on page 8.

11. For parity- there are too many small groups. Please consider collapsing some of the groups eg. 0, 1-2, 3+

Authors’ Response

Done. The parity has been modified as P0, P1-2 and P ≥3. See table 1 and table 4. 

12. There are many small tables - can some of these be combined?

Authors’ Response

Done as shown below. 

Table 2 & 3 merged together as Table 2.

Table 4 & 5 merged together as Table 3 

Table 7 & 8 merged together as Table 5

Hence, we now have a total of 5 tables instead of 8.

13. How was pain assessed? Is this a validated scale?

Authors’ Response

Pain assessment was done using numerical rating scale. NRS have well-documented validity. They correlate positively with other measures of pain and show sensitivity to treatments that are expected to affect pain.

After each smear, the patient was asked to rate her pain at the end of the procedure using a validated numerical rating scale for pain from 0 (no discomfort) to 10 (most discomfort) to indicate if she was willing to come for repeat testing in the future. The numeric rating scale for pain was used because of its authenticity and universal acceptance for the evaluation of pain.11 See line 115 to 118 on page 7.

14. The reason for increased pain in the high parity group is speculated as being due to menopause. Was menopausal status of the women collected? Can you adjust the parity association for menopausal status, or for age? What happens?

Authors’ Response

We appreciate the comments of the reviewers. We did not assess the menopausal status of the participants. We have appended it as part of the limitations of the present study as follows: Also, we did not document the menopausal status of the participants. See line 245 t0 246 on page 12.

15. The trial is described as double blind. The participants surely knew which arm they were allocated to? Similarly, the person performing the pap smear knew which arm the women were allocated to. Please discuss blinding more thoroughly in the manuscript.

Authors’ Response

The primary outcome of the study was proportions of unsatisfactory cervical cytology smears, an outcome that was determined by the histopathologists as one of the outcome assessor, we can state that except the researcher, the histopathologist and participants were blinded and so were not aware of the intervention arm thereby preventing bias and giving more credence to the study. Regarding the participants, the application of the lubricants to the speculum was made when the participants were already in lithotomy or dorsal position with the faces not facing the researchers for collection of specimen. Also, the speculum preparation was made in the other section of the examining room to ensure blinding of the intervention assignment.

We appended under method section as follows:

The primary outcome of the study was proportions of unsatisfactory cervical cytology smears, an outcome that was determined by the histopathologists as one of the outcome assessor, we can state that except the researcher, the histopathologist and participants were blinded and so were not aware of the intervention arm thereby preventing bias and giving more credence to the study. Regarding the participants, the application of the lubricants to the speculum was made when the participants were already in lithotomy or dorsal position with the faces not facing the researchers when collecting the smears. Also, the speculum preparation was made in the other section of the examining room to ensure blinding of the intervention assignment.

See line 97 to 104 on page 7.

 REVIEWER 2 COMMENTS

1. The prevalence quoted in your manuscript is not in tandem with information in the cited reference and therefore not correct. Ca cervix is the 3rd most common female cancer globally and the 2nd in Africa as contained in your referenced publication.

 Authors’ Response

Thank you for the observation. However we rechecked and found out that our initial statement is correct. In terms of incidence, worldwide, Breast cancer is the commonest female malignacy (24.2%), followed by colorectal cancer (9.5%), lung cancer (8.4%) and then cervical cancer (6.6%). With regards to mortality, cervical cancer is responsible for 7.5% mortality from female malignancy worldwide.

Cancer of the cervix is the fourth most prevalent malignancy in females, with a projected incidence of 570,000 in 2018.1 It represents 6.6 percent of all gynaecological carcinomas and is responsible for 7.5 percent of all female malignancy mortality worlwide.1

Ferlay J, Colombet M, Soerjomataram I, Mathers C, Parkin DM. Estimating the global cancer incidence and mortality in 2018: GLOBOCAN sources and methods. Int J cancer. 2019 Apr 15; 144 (8):1941-1953.

2. Methodology: Line 106; Change statement to ... because its authenticit

Authors’ Response

Done. See line 118 on page 7 as shown below.

The numeric rating scale for pain was used because of its authenticity and universal acceptance for the evaluation of pain.11

3. Results: Table 5 is not clear and appears confusing

Authors’ Response

Table 5 compared the pain score of participants in the GEL and NO-GEL group. The mean pain score was significantly higher in the NO-GEL group when compared to GEL group (p<0.001). 

Please note that table 5 and 4 have been merged to become the new Table 3 as requested. See table 3.

Table 3: Comparison of Pain scores of the participants 

Variables Study Group Total (%) p-value

 GEL (%) No GEL (%) 

Numeric rating scale for pain 

(0 = no pain, 10 = worst pain) 1 23 (34.8) 1 (1.5) 24 (18.2) 

 2 13 (19.7) 1 (1.5) 14 (10.6) 

 3 10 (15.2) 0 (0.0) 10 (7.6) 

 4 2 (3.0) 12 (18.2) 14 (10.6) 

 5 6 (9.1) 17 (25.8)

---

## [Editor Report · Decision Letter 1]

30 Apr 2024

Effects of speculum lubrication on cervical smears for cervical cancer screening: a double blind randomized clinical trial

PONE-D-23-28708R1

Dear Dr. OKAFOR,

We’re pleased to inform you that your manuscript has been judged scientifically suitable for publication and will be formally accepted for publication once it meets all outstanding technical requirements.

Kind regards,

David Chibuike Ikwuka, Ph.D.

Academic Editor

PLOS ONE
---

## [Editor Report · Acceptance letter]

14 May 2024

PONE-D-23-28708R1 

PLOS ONE

Dear Dr. Okafor, 

I'm pleased to inform you that your manuscript has been deemed suitable for publication in PLOS ONE. Congratulations! Your manuscript is now being handed over to our production team.

Kind regards, 

on behalf of

Dr David Chibuike Ikwuka 

Academic Editor

PLOS ONE